# Assessing the Integrative Framework for the Implementation of Change in Nursing Practice: Comparative Case Studies in French Hospitals

**DOI:** 10.3390/healthcare10030417

**Published:** 2022-02-23

**Authors:** Israa Salma, Mathias Waelli

**Affiliations:** 1Inserm U 1309-RSMS ARENES UMR 6051, Management Institute, EHESP, CS 74312, CEDEX, 35043 Rennes, France; mathias.waelli@ehesp.fr; 2Global Health Institute, Geneva University, 1202 Geneva, Switzerland

**Keywords:** implementation, integrative framework for implementation of change in nursing practices, strategic level, socio-material factors, interferences, mechanisms of mobilization, leadership, local managers

## Abstract

The implementation of healthcare policies in healthcare organizations is a pivotal issue for managers. They generally require a change in professional practices. In previous work, we developed the Integrative Framework for Implementation of change in Nursing Practices (IFINP) to support implementation initiatives for such change in nursing practices. We aimed to assess the generalizability of IFINP in other organizational settings and explore links between strategic and socio-material factors during implementation. We used a comparative qualitative case study at three French hospitals to assess the implementation of certification procedures. Data were collected from 33 semi-structured interviews with managers and nurses. Narratives reflecting actions and interactions were extracted and deductively analyzed using IFINP components. The results showed that the framework was flexible and captured the different aspects of implementation actions and interactions at the three hospitals. Strong interferences were identified between mobilization mechanisms and strategic elements. Interferences were observed mostly between ‘reflexive monitoring and work articulation’, and ‘reflexive monitoring and sense-making’ mechanisms. Leadership was integrated into the different mechanisms, especially the ‘translation’ mechanism. The IFINP facilitated a greater understanding of strategic elements and associated relationships with social and material factors during implementation. It helps to provide a clear definition of the managers’ role when implementing new nurse practices.

## 1. Introduction

Over the past decades, healthcare policies, and reforms have constantly evolved to improve efficiency and benchmarks for cost-effectiveness and quality of care [1]. Multiple external quality control procedures have been implemented to ensure the quality and safety of patient care [2]. Implementing such quality initiatives is a pivotal issue, first due to the complexity of healthcare systems [3]. Second, healthcare providers often feel disconnected from top-down decisions, as they consider these quality initiatives as being imposed on them [4]. For instance, although quality improvement (QI) initiatives are increasingly adopted in healthcare organizations [5,6], often they lead to sub-optimal outcomes in healthcare [7]. In our study, quality initiatives reflect the structural aspect of quality in patient care, which can be policies, programs, standards, and practice. These create the environment in which functional care processes occur, in our case in nursing activities [8]. Effective implementation of these initiatives is associated with positive patient and staff outcomes and enhances care cost-effectiveness [9]. However, the failure of such an implementation may have a serious impact, causing additional workloads and increased staff burden [10,11]. Previous research reported that anxiety toward integrating innovations in practice is a common concern [12]. Implementing such changes into routine practice is recognized as challenging and its outcomes are unpredictable and uncertain [13]. As a consequence, researchers have investigated how to effectively implement change into clinical practice [14] by identifying factors that may impact implementation processes and using models, theories, and frameworks [13,15].

The implementation, as a subject, has been investigated from two different perspectives. The first draws primarily on a strategic approach. It identifies a wide range of transversal factors and implementation strategies, which can be useful in multiple clinical settings, including leadership, culture, resources, and others [16,17]. Similarly, multiple frameworks, models, and theories have been developed [18,19] to support implementation initiatives [20,21,22]. The second is centered on activity levels and focuses on local socio-material contexts and their impact on implementation processes [23,24]. A socio-material context is built upon the intersection of materials or technologies (e.g., electronic health record EHR, checklists, etc.), work and organization of everyday life. It constitutes the local context of activity in our study nurse’s activity [23]. Additionally, these perspectives focus on a clinical manager’s role to generate quality improvements/results [25]. Andreasson et al. speak of a potential risk of failure in implementing such changes in care processes by top management given the existing gaps between strategic and operational levels in hospitals [26]. Previous research reported that bridging such gaps between strategic and activity levels primarily depends on clinical managers translating and adapting intended changes to local contexts [25,27], as outlined in the theory of middle management roles [28].

Facilitator factors related to work settings at strategic levels [29] and the dynamic aspects of local contexts, and how they interrelate during an implementation process, are both essential steps towards an effective implementation at the activity level [30]. However, these factors are generally, separately addressed. Thus, it is important to address both steps in an integrative framework to identify strategic levels, consider specificity, and analyze the local context of implementation. For example, how is leadership operationalized in an implementation process, given its complexity and the overlapping reality of the local context? Within these perspectives, we previously developed a framework for the implementation of innovation at nurses’ levels, i.e., the Integrative Framework for Implementation in Nursing Practice (IFINP) [31]. The framework was developed using a two-step mixed approach. The first was an inductive analysis of the certification implementation procedure at a teaching hospital. The second was a deductive analysis using two theoretical approaches: the quality implementation tool (QIT) [32] and translational mobilization theory (TMT) [33].

While multiple implementation frameworks have been developed, limited studies have evaluated the usefulness of these promising approaches [18]. In this study, we first tested the IFINP using certification implementation procedures as an example in multiple case study settings (three French hospitals). Our strategy was to identify how framework components captured implementation processes in multiple organizational settings. Secondly, we explored the links between strategic elements and socio-material factors of the implementation process in a local context.

## 2. Materials and Methods

### 2.1. Choice of Certification Procedures

The implementation of certification procedures was used as the subject to test the IFINP. First, a certification is considered as one of the main “peer evaluation techniques” in Europe, which is based on the International Organization for Standardization. In France, certification is mandatory in both public and private health organizations [34]. Thus, it was useful to study implementation processes in different organization types. Second, the evaluation strategies of certification are based on standards and benchmarking, and must, given this, include the best clinical practices, processes audits and associated quality and safety indicators [34]. The implementation of certification regulations is a key managerial issue, in terms of its integration and sustainability in professional routine practices [35], especially, where it is perceived, given the multiple requirements and increasing workload for professionals [6], mainly primarily nurses [36]. However, the requirements are essential for improving care quality and patient safety [37]. Thus, it was important to assess the framework utility in understanding the implementation processes of certification implementation procedures in multiple organizational settings.

### 2.2. Study Design

We used a comparative qualitative case study approach. A case study examines phenomena in ‘real life’ contexts [38], e.g., understanding the implementation of an intervention in a healthcare system [39]. This approach explores phenomena from different perspectives; “Through case-by-case comparisons, the analyst fine-tunes, modifies, and qualifies the propositions so that they express precisely the limiting conditions revealed by the pattern of findings across all cases” [40].

### 2.3. Study Locations

In order to assess the IFINP in multiple contextual settings, we selected two hospitals in western France, distinguished by size, type, and status. IFINP had previously been developed by us in high-risk sectors at a teaching hospital center (hospital A) [31]. To broaden our remit, we investigated other sector types in hospital A, and also other hospital types (B and C) (Table 1). Thus, this study was based on a sample of three hospitals only. Given the COVID-19 pandemic, multiple hospitals refused to participate.

### 2.4. Data Collection

Data was collected using semi-structured interviews with relevant actors involved in certification implementation procedures (Figure 1). Due to the ongoing COVID-19 pandemic and associated restrictions, we were unable to conduct observations at study sites.

#### Interviews

As multiple actors at different organizational levels were involved in certification procedures, interviews were conducted with actors at different hierarchies. This approach provided in-depth insights on the role and responsibilities of actors in each local context and provided a better understanding of the different factors impacting the process [41]. Interviews were conducted until ‘data-saturation’; interviews were conducted until outputs provided non-essential data related to study objectives [42]. To avoid bias related to directed enquiries on framework components, interviews were conducted in a comprehensive manner; we discussed the implementation of certification procedures and processes at nurse activity levels. Furthermore, we elaborated on elements contributing to successful change integration processes imposed by these procedures on nurse practices. Interviews were conducted by the PI (IS), either face-to-face or online, according to hospital regulations and participant preference.

In total, 33 semi-structured interviews were conducted at the 3 sites. To ensure participant anonymity, interviews were sequentially numbered in each site, using acronyms based on participant roles: TL, top leader; MM, mid-manager; and RN, registered nurse. A, B, and C denoted the site. For example, the top leader 1 at site B is TL1-B; register nurse 5 at site C is RN5-C.

### 2.5. Data Analysis

#### 2.5.1. The Theoretical Framework

The IFINP framework was developed to conceptualize innovation implementation at nursing activity levels in French hospitals (Figure 2). The framework distinguished two key components in implementation certification procedures. Firstly, contextual settings were considered as strategic elements, e.g., actors, organizational logistics, structures, materials, technologies, interpretative repertoires, and implementation leadership approaches [43]. Secondly, mobilization mechanisms encompassed actions, practices, and interactions between these elements. The framework incorporated five mechanisms: (1) object formation, (2) translation, (3) sense-making, (4) reflexive monitoring, and (5) work articulation. These shaped and guided the implementation processes, thereby reflecting local socio-material factors [31]. The framework also showed how an implementation context consisted of both social and material elements interacting together in a continuum, rather than a linear ‘pipeline’ approach [44].

IFINP (Figure 2) facilitates the implementation of innovation into practice. It identifies different macro and meso levels during an implementation process. Macro levels reflect healthcare systems. Meso levels reflect organizational levels that consist of contextual settings and actors involved in certificate implementation processes at different organizational levels. Mobilization mechanisms also include object formation, translation, work articulation, reflexive monitoring, and sense-making. These shape the interrelationships between framework components. IFINP also identifies the leadership approach of change leaders at local levels (champions and/or local managers) [31].

#### 2.5.2. Data Coding

All narratives reflecting implementation processes, such as actions, interactions, key factors, contextual settings, and others were stored in separate computer files with respect to each hospital. Narratives were then used in a deductive analysis using framework elements in a table format (Table A1 (Appendix B)). To ensure analytical credibility, both authors conducted a simple test to characterize categories and define inclusion and exclusion criteria; authors separately conducted coding for a sample of narratives (*n* = 30) according to definitions (Table A1). Then, a discussion followed regarding the primary results of coded sample narratives. This process helped frame each category and define inclusion and exclusion criteria. Then, the principal investigator (IS) performed narrative coding steps. Study reporting guidelines were based on consolidated criteria for reporting qualitative research (COREQ) (Appendix A) [45].

### 2.6. Research Ethics

This study involved professionals and no patients or human experiments. In France, this type of study does not require institutional review board (IRB) authorization, as is the case in the United States [46]. In France, according to “Jarde law” L1121-1 PHC, three study types involving humans require ethical approval: (1) human interventional studies, (2) studies with minimal risk and intervention, (3) and non-interventional studies (in the context of patient data) [47]. According to qualitative research ethics guidelines [48], signed consent from participants is adequate, and interviews should be conducted in private, comfortable, and informal settings. In this study, participants were free to participate. Furthermore, to maintain strict anonymity, interviewees, and interview transcripts were anonymized and assigned acronyms.

## 3. Results

Firstly, all interviewee’s narratives reflecting actions and elements in certification implementation procedures were captured by the framework (Table 2). IFINP categories described emergent issues at all sites (Cases A, B, and C) and sectors in the same hospital, whether general medicine, ICU, and interventional sectors (endoscopy and the operating room). Thus, the framework recognized the mobilized implementation elements, actions, and interactions for the implementation of certification procedures into routine practice.

Secondly, we identified overlapping aspects with respect to framework elements. Multiple narratives showed interference between two or more mobilization mechanisms and also between strategic elements and mobilization mechanisms (Table A2 (Appendix C)). Interference is understood as an overlap or an intrusion of two or more elements during a process. In this study, we used ‘interference’ to describe the overlaps identified in certification implementation procedures. Interview analysis showed that overlaps were mostly observed between both ‘reflexive monitoring and work articulation’ and ‘reflexive monitoring and sense-making’ mechanisms at the three sites. However, the ‘object formation’ mechanism was only weakly associated with the ‘translation’ mechanism when compared to the other mechanisms. Some narratives reflected associations between multiple mechanisms, and sometimes with contextual elements. For example, *“we use quality meetings to explain certification procedures to managers and professionals. We work with the quality unit who provide regular updates on different indicators and outcomes. Also, our auditing systems help us monitor the integration of new procedures into routine practice” TL1-C* reflecting shifts between ‘translation’, ‘reflexive monitoring’, and ‘work articulation’. In another example, *“for implementation procedures, we identify referents/champions, we improve their skills and train them in methods and tools required for certification, quality and risk management, so they can introduce/implement change and help nurses to change” TL2-B.* This statement reflected the leadership of the referent/champion through ‘reflexive monitoring’ and ‘work articulation’ mechanisms. In this study, ‘referent’ is a key actor in the implementation process and is considered a champion; the term refers to their role [31]. In addition, leadership was associated with each mechanism but strongly interfered with the ‘translation’ mechanism. We identified two leadership levels. The higher level generally interfered with ‘object formation’ mechanisms; *“we are supported by the quality unit of our hospital in implementing quality policies; their help defines the working plan at different levels” MM2-A.* The leadership proximity manager and/or the leader of change at the professional level states, *“we provide necessary training for nurses and we consider different contextual settings in which to implement new practice changes, such as working procedures and essential documents, so when we introduce change we have everything in place” MM2-B.* Given the dynamic aspect of an implementation context, it is important to consider the local socio-material impact as well as barriers and facilitators that may impact implementation processes. This operationalizes strategic elements, such as leadership within a local context and shows how they are interrelated (Figure 3).

The following sections outline stratified comparisons between the three case studies based on interferences identified between mobilization mechanisms and leadership.

### 3.1. Comparison of ‘Object Formation’ and ‘Translation’ Mechanisms and Interferences with Leadership

‘Object formation’ reflects the initial mobilization step of certification procedures within an organization. We identified similar practices for this initial step across the three sites (Table A3 (Appendix D)). Generally, the TL, MM, specialists, and a steering committee enacted action plans based on certification criteria and departmental evaluations. This plan defined the objectives and actions of each department. All information regarding procedures or actions was communicated by managers or informatics systems. At the nurse level, referents/champions or local managers were charged with communicating procedures and changes, either during meetings, e-mail and/or circulated documentation. In essence, they prepared the local context to accept change. ‘Object formation’ was accompanied by ‘translation’ mechanisms. The introduction of any procedure must be defined in terms of needs and requirements. For example, at the organizational level, this was done by explaining ‘why’ and ‘how’ a procedure was to be introduced and integrated at each level. At the local professional level, this was justified as a procedural need and benefit for patient care and depended on the leadership skills of proximity managers to use formal and informal strategies to support and provide meaning to the implemented change.

### 3.2. Comparison of ‘Sense-Making’, ‘Reflexive Monitoring’, and ‘Work Articulation’ Mechanisms with the Leadership

Communicating information and explaining procedures does not guarantee an effective implementation; it must also make sense for professionals: *“we can write a procedure, we can explain it, and we can introduce it to nurses, but the most important thing is that this procedure is effectively implemented in their daily practice” MM2-A.* We identified multiple managerial strategies to make sense of the implemented procedures in nurse practices (Table A4 (Appendix E)). Primarily, managers insisted on involving nurses in these procedures, often at inception. Nurses were actively engaged in the development and writing of procedures (sites A and B). This also involved organizing interventions, giving feedback and opinions on procedures and changes (site C). Local managers also provided administrative support for professionals using informatics systems or documentation. These steps were essential for working procedures and policies. This was identified as a tool supporting professionals’ practices. In addition, to effectively implement a procedure, interviewees highlighted the usefulness of a pilot period, which allowed professionals to live and experience the change, readjust and adapt according to local context reality, and unlimitedly accept it and use it.

We observed similarities in ‘monitoring’ and evaluation methods across the three sites. ‘Reflexive monitoring’ was represented by ‘formal’ tools, such as auditing systems, indicators, professional practice evaluations, and adverse events. Additionally, ‘informal’ methods were facilitated through a professional’s feedback on procedural feasibility, which relied heavily on the local leadership. Nurses provided feedback either directly to local managers and referents, or at regular team meetings. This helped evaluate procedures and air concerns. The ‘reflexive monitoring’ mechanism interfered not only with ‘sense-making’ mechanisms, but also with ‘work articulation’ mechanisms. Continuous monitoring was fundamental to ensuring corrective actions and improved procedural integration. These continuous improvements are incorporated into work articulation mechanisms. The ability to conduct continuous and regular meetings and ensure communications between actors at multiple organizational levels allowed actors to readjust, adapt, and formalize change trajectories.

## 4. Discussion

We tested our IFINP framework for the implementation of certification procedures using a comparative case study design in three French hospitals.

Firstly, we demonstrated framework flexibility in capturing the reality of certification procedure implementation in multiple settings in different French healthcare organizations. The IFINP successfully identified different actions and interactions between actors, contexts, and implemented procedures, regardless of sector type, hospital type, and size. Although the framework was originally constructed using a case study in high-risk sectors at a teaching hospital [31], it was practical for explaining certification procedures in other sectors (medicine; case A), as well as other hospital types (cases B and C). This provided a formal framework to understand mechanisms where individual and organizational contexts affected innovation integration into nursing practice [18].

Secondly, we revealed strong interferences between the IFINP elements during implementation processes. Repeatedly, participant narratives reflected an interposition between different mobilization mechanisms during certification implementation across the three sites. However, mostly, interferences were identified between ‘object formation’ and ‘translation’ mechanisms at higher manager levels, and ‘sense-making’, ‘reflexive monitoring’, and ‘work articulation’ mechanisms at activity levels. This may be explained by the presence of two implementation phases. The first phase reflected the adoption of certification procedures at the organizational level and involved actions related to the preparation of the initial mobilization and change diffusion, and was mostly seen at higher levels. For instance, through regular teams’ meetings, leaders defined the organizations’ plan with other actors, and they explained and translated certification criteria in daily practice.

The second phase reflected the appropriation of implemented change by professionals into their routine practice. This involved different actions by local managers, or change leaders, leading to the effective integration of change. For example, professional involvement in procedures responded to ‘sense-making’ mechanisms and was observed by an active engagement via analyses and evaluations. Furthermore, feedback and improvement suggestions interfered with the ‘reflexive monitoring’ mechanism. Additionally, through ‘reflexive monitoring’, e.g., monitoring meetings, managers, and professionals defined corrective actions, they continuously evaluated, adapted, and readjusted implemented procedures according to local context requirements and interfered with the ‘work articulation’ mechanism. The healthcare system complexity [49] is accompanied by implementation procedural complexity—via multiple contributors and multifaceted and multidimensional strategies [50]. This situation requires a dynamic constituent to improve the uptake of important changes by professionals [51]. Thus, identifying interferences in IFINP mobilization mechanisms and elements during certification implementation procedures is important and supports the non-linearity aspect of implementation processes [24].

In addition, the IFINP helped exemplify the leadership factor. Leadership interference with different mobilization mechanisms was useful in defining the content and activity undertaken by change leaders and their response to mobilization mechanisms. For example, leadership at the organizational level involved top leaders providing information and clear instructions on adopted changes. They also supported managers and professionals at regular meetings [52,53]. This situation reflected an interference of leadership in terms of ‘object formation’ and ‘translation’ mechanisms. In addition, leadership approaches [54] at the local level interfered with translation, sense-making, reflexive monitoring, and work articulation mechanisms. This scenario provided important insights into the change leaders’ role in translating and adapting procedures to the local context and thus, integrating them into professional practice. These outputs also highlighted their willingness to implement certification procedures at the three sites. From this, a question arose on the place of local managers’ roles and activities, which must be considered by decision-makers in implementing quality policies [25]. Using the IFINP, we showed that the leadership approach involved considerable translation, support, and monitoring changes, whereas other strategic approaches emphasized the leadership as facilitators, without clear conceptualization [55,56]. Thus, the IFINP helped frame these elements within the local implementation context.

Thirdly, the stratified comparisons of IFINP elements based on overlaps revealed major similarities between implementation strategies and interventions across different sector types (case A) and hospitals (cases B and C). This suggested an independent aspect of certification implementation strategies in terms of multiple sectors’ types at the three sites. This could be explained as the generation of work harmonization and standardization processes between French healthcare organizations in terms of quality management [57] and which can be understood by the concept of institutional isomorphism [58]. DiMaggio and Powell (1983) identified three mechanisms: coercive, mimetic, and normative, through which isomorphism occurs [59]. A certification responds to coercive and normative mechanisms. First, it is performed by external stakeholders to implement different regulations and standards [60]. Second, it emphasizes the need for professionalization and the importance of establishing cognitive and professional standards [59]. Thus, we suggest that implementing such changes in nursing practice is established within the framework of institutional isomorphism for quality improvement in patient care [61,62]. To address this suggestion, we advocate the study of international contexts and other types of managerial innovation.

### Study Limitations

We acknowledge several study limitations. Our cases focused on the implementation of certification procedures at the nurse level, thus participants at the activity level were nurses. However, the implementation scope was broad and certification procedures involved many professionals, not only nurses. Thus, we may have missed data on some implementation processes. Secondly, in terms of our data collection methods, the study was primarily based on semi-structured interviews. However, given the COVID-19 crisis and associated hospital restrictions, a limited number of hospitals participated in this study, and we were unable to conduct observations at sites, which may have limited some study elements. However, to overcome this, we thoroughly discussed certification implementation procedures with actors and sought examples from their previous experiences. Finally, we tested the framework in multiple hospital settings, but based on the last iteration of certification and not during the implementation process of certification. However, in the future, it will be interesting to test the IFINP in real-time of certification procedures implementation using on-site observations and interviews in a larger sample of hospitals and sectors.

## 5. Conclusions

Using comparative cases studies, we assessed the IFINP. The framework robustly provided insights on existing interferences between framework components, mechanisms, and elements, and practically explained certification implementation procedures in multiple contexts. The IFINP provided concrete and explicit insights on leadership in terms of a change in a leader’s activity in the implementation process. Our framework goes beyond previous work in implementation by offering an integrated perspective on the social and material elements of the local context involved in implementation processes. Therefore, we advocate IFINP as a tool for managers and policymakers to support the implementation of quality initiatives in nursing practices, regardless of the organization type.

## Figures and Tables

**Figure 1 healthcare-10-00417-f001:**
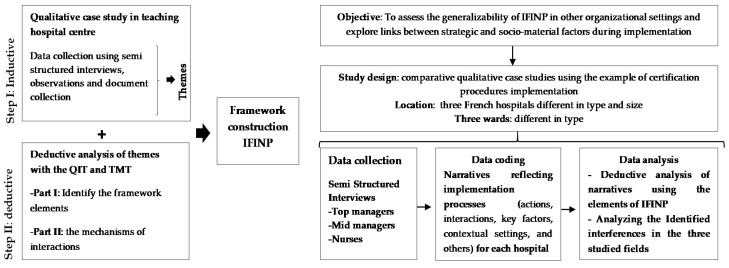
Flow diagram summarizing the used methodology to construct and assess the IFINP.

**Figure 2 healthcare-10-00417-f002:**
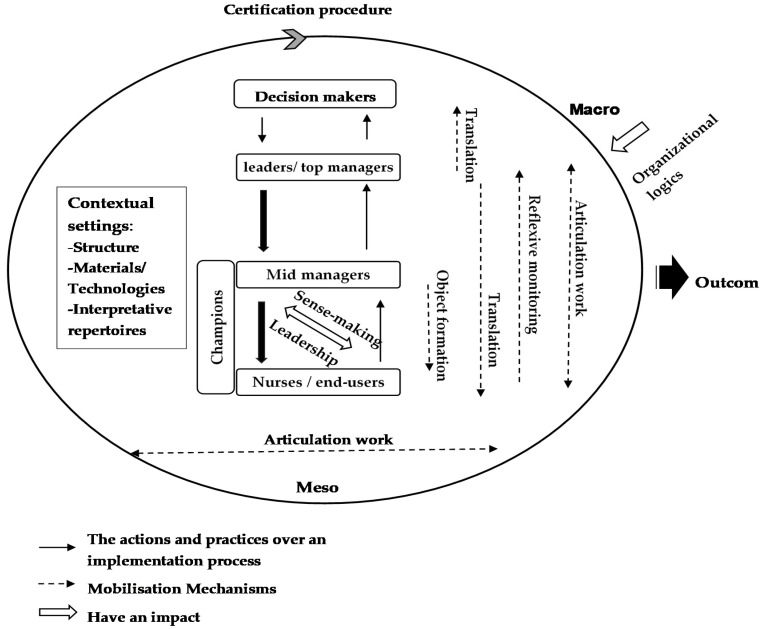
The Integrative Framework for Implementation of change in Nursing Practices (IFINP).

**Figure 3 healthcare-10-00417-f003:**
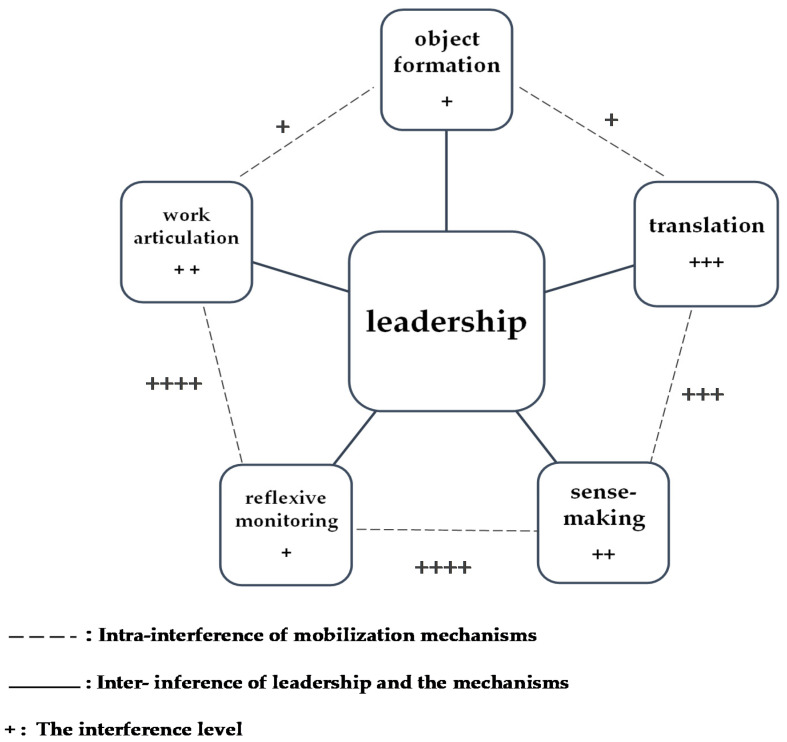
General representation of interference on mobilization mechanisms and leadership elements. The + signs represent interference levels, the power of interferences seen is reflected by the number of + signs such as ++, +++ or ++++. Dashed lines represent intra-interference mechanisms. Full lines represent the inter-interference of leadership and mechanisms.

**Table 1 healthcare-10-00417-t001:** Hospital characteristics.

Hospitals	A	B	C
Type	Teaching hospital	Hospital	Hospital
Size (beds)	924	991	450
Status	Public	Public	Private
Selected Sites	Medicine	Medicine Intensive care unit (ICU) Endoscopy	Medicine Palliative care Operating room

**Table 2 healthcare-10-00417-t002:** Analysis of the three studied contexts using the IFINP components.

Elements		A	B	C
Mechanisms of Mobilization	Object formation	‘often it is our manager that alerts us to a change in protocol’ RN2	‘We prepare our action map according to certification requirements. Also, all the identified risks are objectified and we define our corrective actions. These are integrated into our quality care action plan’ TL1	‘We put the new document on the online document management system, in order to be accessible for all professionals. We diffuse an information that it is implemented. Then each local manager is responsible to diffuse the information to their teams and implement the document’ TL2
	Translation	‘As a local manager we are regularly obliged clarify the interest of new procedure to professionals, why we do it, for what purpose. It is not because we write or adapt the procedure to service it will be implemented!’ MM2	‘we have to explain for nurses that, what they are doing in terms of certification procedures is beneficial for patient care and to improve their work, even if it is perceived as additional traceability or work’ MM1	‘we have to clarify that the new procedure has an interest for them and for the patient, they must find a benefit which will help change their habits a little’ MM2
	Sense-making	‘Nurse are involved in the implementation process. In fact, I can’t do it alone, because I don’t know all about their daily difficulties. I think they will be much more precise in the finesse of things, that it is why they must be engaged’ MM1	‘The fact that we are not directly imposing a solution but involving them (nurses) in the debate during the preparations for implementation, is major facilitator to integrate changes into their routine, I think’ MM2	‘In fact to write a procedure with professionals can guarantees a better appropriation. For example, bring them to reflect on their practice and work with us on the improvement possibilities gives sense to their practices’ TL2
	Reflexive monitoring	‘For a new protocol we have to adapt it and use it. Once we get used to it, we evaluate after that we readjust, readapt and reevaluate what is blocking or the things that are not coherent’ RN1	‘at times we will have some lack, one of things that we are going to implement do not necessarily fully integrated. The feedback of services will alert us on problem. And sharing professional experience and feedback to enrich services on others previous experience, so that they do not relive the same problem’ TL2	‘we have to report a malfunction in terms of the implemented changes, and also questioning the quality department, so this implemented changer can be readjusted’ RN1
	Work articulation	‘sometimes we have to go to training to learn gestures or understand why we make a gesture in such and such a way, here we discuss between us about the new change and also we exchange information’ RN2	‘every week there is a staff meeting in which we explain, observe, evaluate and analyze, so that teams can appropriate more’ MM1	‘the quality department analyzes and then following the degree of feedback, we can organize a meetings to point out the concerns that we encounter to adjust’ RN1
Contextual elements	Organizational logics	‘Really it depends on an organizational culture of quality and patient safety, it’s all in that spirit’ MM2	‘I think it’s a culture, the Culture of improving care facilitates the implementation of certification procedures’ MM1	‘we have to boost the culture of the quality approach between professional, which is quality and risk management culture’ TL1
	Structure	‘We are supported by the quality unit for the implementation of quality policies. The unit defines the working plan at different levels’ MM2	‘we have to create a steering committees with all the departments, all the wards heads, the pole managers to be able to discuss all the themes in order to start organization’, TL2	‘there is members of the management committee or wards executives, thematic referents, different bodies the CLIN * the CLUD *, we have professionals who can be nurses or other professionals’ TL2
	Materials and technologies	‘first, we must have the materials in our disposal, which is it necessary to implement a new procedure’ RN2	‘We conduct always an analysis of the situation, we review we have and potential resources that we can have, and also we work with the concerned people’ MM2	‘Usually the procedure is created, often it is by a higher level it means the direction. we have our informatics system in which all our protocols are grouped together’ RN5
	Interpretative repertoire	‘For example we have a protocol file in the department, in which is identified how to conduct a such and such care, it means the working process of care that should be followed’ RN2	‘We already have tools supporting the implemented changes. For example, on the computer there is a folder for the recent information, we also have an information file. I use these sometimes for certain protocols’ RN1	‘We have an administrative support for our protocol, and we know that we can refer for information in there. I think, this, helps a lot, not only to go have all the information supporting our practice but also to be up to date’ RN5
	Implementation leadership	‘The proximity manager it has a central role in the appropriation of caregivers to change, by their functioning mode. as proximity manager, I think I am really in the loop, we go within the teams and we identify main elements and barriers, and we try to find solutions’ MM1	‘We support them (nurses) on their knowledge and competence, their own current resource, In fact we listen to their need for supervision, and support then on their own practice’ MM2	‘I am there in pilot of certification. I actually organize the dispatching of certification themes of different actors, and I ensure the proper follow-up and the good timing with the other pilots in charge of the in implementation at the activity level’ TL2
	Champions	‘The nurse ‘referent’ participates in the implementation process in the concretization in the drafting of the quality approach, she can also give ideas, but this is more by the quality unit and managers’ MM2	‘I was hygiene referent, I was like an interlocutor of the hygiene cell of the hospital, in fact as hygiene referent I have lot of organizing role, for example when the hygiene protocols change we informed the team, put the change in file of information’ RN1	‘but all nurses are concerned in the implementation of certification procedure, however you have motor nurses who are generally the specialist referents and then others who follow more or less voluntarily’ TL1

CLIN *: nosocomial infection control committee; CLUD *: committee for pain relief and control.

## Data Availability

Datasets (which include individual transcripts) are not publicly available due to confidentiality policies. However, they may be obtained from the corresponding author upon reasonable request.

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
