# Peer review of "Assessing the Integrative Framework for the Implementation of Change in Nursing Practice: Comparative Case Studies in French Hospitals"

_healthcare, 2022, doi:10.3390/healthcare10030417_

Round 1

Reviewer 1 Report

In this work, authors tested their IFINP framework for the implementation of certification procedure using comparative case studies in three French hospitals. The topic is original, as in the previous article they developed their integrative framework and current study is focused on its application. 

I have the following suggestions to be solved:

- Line 13 - ". we aimed to " correct to "We aimed to" - and correct similar imperfections through the text,

- Abstract have to be rewritten as there are "Background, Conclusion, Methods",

- Insert short paragraph what is new and what is known regarding your research, and why is your approach better than existing methods - it could be also written in the conclusion section,

- regarding methodology, maybe some flow chart will be appropriate, as you can show the steps from the previous article - development of framework to the testing and its results.

- please add into the Introduction some recently written papers in relation to this issue.

- please improve your figures, some of them are not easily readable, e.g., Fig. 1,

-  the caption of Figure 1 should be written as a part of the text, not in this caption,

- Fig. 2 - the first letter should be capitalized, - some sub-sections are not numbered, as in section 4. If not, please specify.

Author Response

Response to Reviewer 1 comments

Thank you for reviewing our manuscript titled “Assessing the Integrative Framework for the implementation of change in Nursing Practice: comparative case studies in French hospitals” and submitted to Healthcare Journal. We appreciate the time and effort that you have dedicated to providing your valuable feedback on the manuscript. We are grateful for the insightful comments on the paper. We are proposing a response to all the comments provided which are clarified below and the manuscript has been revised accordingly. The revisions made to the manuscript were marked up using the “Track Changes

Comments and Suggestions for Authors

In this work, authors tested their IFINP framework for the implementation of certification procedure using comparative case studies in three French hospitals. The topic is original, as in the previous article they developed their integrative framework and current study is focused on its application. 
I have the following suggestions to be solved:

Point 1: Line 13 - ". we aimed to " correct to "We aimed to" - and correct similar imperfections through the text

Answer: correction was made. 

Point 2: Abstract have to be rewritten as there are "Background, Conclusion, Methods",

Answer: correction was made. 

Point 3: Insert short paragraph what is new and what is known regarding your research, and why is your approach better than existing methods - it could be also written in the conclusion section.

Answer: thank you for the comment we added more details on the conclusion section. We hope this change meets the expectation of reviewers (page 11, lines: 397-399).

5. Conclusions

Using comparative cases studies we assessed the IFINP. The framework robustly provided insights on existing interferences between framework components, mechanisms and elements, and practically explained certification implementation procedures in multiple contexts. The IFINP provided concrete and explicit insights on the leadership in terms of a change leader’s activity on the implementation process. Our framework goes beyond previous work in implementation by offering an integrated perspective for the social and material elements of local context involved in implementation processes. Therefore, we advocate IFINP as tool for managers and policy makers to support the implementation of quality initiatives in nursing practices, regardless of the organization type.”

Point 4: regarding methodology, maybe some flow chart will be appropriate, as you can show the steps from the previous article - development of framework to the testing and its results.

Answer: thank you for the suggestion we added a flow diagram to the methodology section. We hope this change meets the expectation of reviewers (page 3, lines: 117 Figure 1).

Point 5: please add into the Introduction some recently written papers in relation to this issue.

Answer: thank you for the comment, we would like to point out that some of our references were ancient due to the era when the topic of quality improvement and implementation science took interest in research. However, we have added more recent references in the introduction section related to the studied subject. We hope this clarification and the modification we made meet the expectation of reviewers.

References: 2, 8, 11 and 22.

Point 6: please improve your figures, some of them are not easily readable, e.g., Fig. 1

Answer: correction was made, and currently Fig. 1 became Fig. 2 (Page 5)

Point 7: the caption of Figure 1 should be written as a part of the text, not in this caption,

Answer: correction was made. The caption was integrated in the Theoretical framework section (page 4, 5; Lines: 178-185)

Point 8: Fig. 2 - the first letter should be capitalized, - some sub-sections are not numbered, as in section 4. If not, please specify.

Answer: correction was made for the Fig.3 (currently).

Regarding the discussion subsections, thank you for the suggestion, but we feel that the discussion section should be as an entire section discussing all the findings following the logical order of the paper. This in order to avoid any separation and disconnection between main findings for readers

We hope that the new version of our paper meets the expectations of the Editorial Board of Healthcare Journal. Please do not hesitate to contact us at your convenience if you need further information. 

 Yours sincerely,

Miss Israa SALMA

Reviewer 2 Report

This is a very interesting paper on the mechanisms and processes that facilitate the integration of strategies with operations to improve the quality of care in three hospitals in France.   The logic of presentation is sound and adequately describes the need for performing a qualitative study on nursing practice and leadership.  Although it is a very readable paper, several improvements are suggested as follows:

  1. Definition or Operationalization of Terms Used:  Many terms are not adequately defined in the introduction section.  For instance, material aspects are defined in the table shown in the methods section.  If it is not defined clearly, we could assume "material"  to refer to "resources".
  2. Structural vs. Process Aspects of Quality:  It is very confusing if they are not clearly operationalized in this study.  Do you mean the structural design of the nursing practice?  Authors are encouraged to review previous publications on nursing systems research published in the Western Journal of Nursing Research or Nursing Research.
  3. Institutional Isomorphism:  Nursing professional practice has been established within the framework of institutional isomorphism for quality improvement.  Authors should carefully cite relevant studies on how the three mechanisms (i.e., coercive, normative, and mimic mechanisms) in achieving an optimal QI in nursing practice and patient outcomes.  Certification could be considered as a coercive and/or normative mechanism).
  4. Sampling Hospitals Studied:  Please explain the limitations of the purposive sample (N=3) used.  How can you improve your sampling framework and design in future research?
  5. Future Research:  It would be a good idea to show how the design limitations could be resolved in a more scientific research design.

Author Response

Response to Reviewer 2 comments

Thank you for reviewing our manuscript titled “Assessing the Integrative Framework for the implementation of change in Nursing Practice: comparative case studies in French hospitals” and submitted to Healthcare Journal. We appreciate the time and effort that you have dedicated to providing your valuable feedback on the manuscript. We are grateful for the insightful comments on the paper. We are proposing a response to all the comments provided which are clarified below and the manuscript has been revised accordingly. The revisions made to the manuscript were marked up using the “Track Changes

Comments and Suggestions for Authors

This is a very interesting paper on the mechanisms and processes that facilitate the integration of strategies with operations to improve the quality of care in three hospitals in France. The logic of presentation is sound and adequately describes the need for performing a qualitative study on nursing practice and leadership.  Although it is a very readable paper, several improvements are suggested as follows:

Point 1: Definition or Operationalization of Terms Used: Many terms are not adequately defined in the introduction section.  For instance, material aspects are defined in the table shown in the methods section.  If it is not defined clearly, we could assume "material" to refer to "resources".

Answer: thank you for the comment we elaborated more on the subject of “social and material” in the introduction by adding a definition and some examples.  We hope the changes we made meet the expectation of reviewers (page 2, lines: 59-62).

“The second, is centered on activity levels and focuses on local socio-material con-texts and its impact on implementation processes [23,24]. A socio-material context is built upon the intersection of materials or technologies (e.g. electronic health record EHR, checklists…), work and organization of everyday life. It constitutes the local context of activity, in our study nurse’s activity [23]”

Point 2: Structural vs. Process Aspects of Quality: It is very confusing if they are not clearly operationalized in this study.  Do you mean the structural design of the nursing practice?  Authors are encouraged to review previous publications on nursing systems research published in the Western Journal of Nursing Research or Nursing Research.

Answer: thank you for the suggestion, to avoid confusion we added into the introduction suction more details on the studied aspects of quality. We hope these modifications meet the expectation of reviewers (page 1, lines: 41-43).

“For instance, although quality improvement (QI) initiatives are increasingly adopted in healthcare organizations [5,6]. Often they lead to sub-optimal outcomes in healthcare [7]. In our study, quality initiatives reflect the structural aspect of quality in patient care, which can be policies, programs, standards, and practices creating an environment in which functional care processes, in our case in the nursing activities [8]”

Point 3: Institutional Isomorphism: Nursing professional practice has been established within the framework of institutional isomorphism for quality improvement.  Authors should carefully cite relevant studies on how the three mechanisms (i.e., coercive, normative, and mimic mechanisms) in achieving an optimal QI in nursing practice and patient outcomes.  Certification could be considered as a coercive and/or normative mechanism).

Answer: thank you for the comments and this suggestion, we elaborated on the subject of Institutional Isomorphism in the discussion section. We hope the added part meets the expectation of reviewers (page 11, lines: 367-347).

“This could be explained as the generation of work harmonization and standardization processes between French healthcare organizations in terms of quality management [57], and which can be understood by the concept of institutional isomorphism [58]. DiMaggio and Powell (1983) identified three mechanisms: coercive, mimetic and normative, through which isomorphism occurs [59]. A certification responds to a coercive and normative mechanisms. First, because it performed by external stakeholders to implement different regulations and standards [60]. Second, as it emphasizes the need for professionalization and the importance for establishing cognitive and professional standards [59]. Thus, we suggest that implementing such changes in nursing practice is established within the framework of institutional isomorphism for quality improvement in patient care [61,62].

Point 4 : Sampling Hospitals Studied: Please explain the limitations of the purposive sample (N=3) used.  How can you improve your sampling framework and design in future research?

And

Point 5: Future Research: It would be a good idea to show how the design limitations could be resolved in a more scientific research design.

Answer: thank you for the comment. We clarified the reason of the limitations in the hospitals sampling. Also we added in the limitations section how to improve in a future research. We hope these modifications meet the expectation of reviewers (Page 3, lines: 138-139 and page: 11, lines: 383, 390,391).

We hope that the new version of our paper meets the expectations of the Editorial Board of Healthcare Journal. Please do not hesitate to contact us at your convenience if you need further information.

Yours sincerely,

Miss Israa SALMA